# Prolonged Consumption of A2 β-Casein Milk Reduces Symptoms Compared to A1 and A2 β-Casein Milk in Lactose Maldigesters: A Two-Week Adaptation Study

**DOI:** 10.3390/nu16121963

**Published:** 2024-06-20

**Authors:** Monica Ramakrishnan, Sindusha Mysore Saiprasad, Dennis A. Savaiano

**Affiliations:** Department of Nutrition Science, College of Health and Human Sciences, Purdue University, West Lafayette, IN 47907, USA; smysores@purdue.edu (S.M.S.); savaiano@purdue.edu (D.A.S.)

**Keywords:** A1 β-casein, A2 β-casein, A2 milk, lactose maldigestion, inflammation, lactose intolerance, betacasomorphin-7

## Abstract

Approximately 30% of milk protein is β-casein. We aimed to determine whether lactose maldigesters who chronically consumed two cups of A1/A2 milk (containing 75% A1 β-casein and 25% A2 β-casein) would adapt to have fewer intolerance symptoms, lower serum inflammatory markers, and/or altered glutathione levels similar to those consuming A2 milk (containing 100% A2 β-casein). A double-blinded, randomized, crossover trial was conducted. Sixteen confirmed lactose maldigesters consumed 250 mL of A1/A2 milk and A2 milk twice daily with meals for two weeks. At the end of the adaptation period on day 15, lactose maldigestion was measured after a challenge with the same milk used for adaptation (0.5 g of lactose per kg of body weight) with a hydrogen breath test. Fecal urgency was higher during the two-week consumption of A1/A2 milk compared to A2 milk (*p* = 0.04, *n* = 16). Bloating (*p* = 0.03, *n* = 16) and flatulence (*p* = 0.02, *n* = 16) were also higher on the 15th day with A1/A2 milk compared to A2 milk challenge. However, day-to-day symptoms, hydrogen, serum inflammatory markers, and antioxidant concentrations were not different after A1/A2 and A2 milk consumption adaptation periods. Adaptation over two weeks did not improve lactose digestion or tolerance of A1/A2 milk to match that of A2 milk.

## 1. Introduction

Lactase, the enzyme responsible for lactose digestion, exhibits high expression levels during infancy and early childhood and decreases following weaning in two-thirds of the world’s population [1,2]. These lactase non-persistent individuals maintain reduced lactase expression during adulthood [3]. An additional critical aspect of lactose maldigestion may be mucosal inflammation. Inflammation can lead to mucosal damage and lower enzyme expression and contribute to lactose maldigestion [4,5].

Approximately 30% of milk protein is β-casein [6]. The influence of two β-casein variants, A1 and A2 β-casein, on lactose intolerance has been extensively investigated [7]. A2 β-casein, the ‘old world variant’, differs from the ‘new world variant’ A1 β-casein due to a single-nucleotide polymorphism (SNP) that replaced proline at the 67th position with histidine [8]. A1 β-casein, containing histidine, causes the release of the pro-inflammatory compound betacasomorphin-7 (BCM-7) during digestion, whereas A2 β-casein, with proline, does not release BCM-7 [9,10].

Previous studies have shown that milk containing A1 β-casein leads to more pronounced lactose intolerance symptoms [11,12,13,14]. Our previous research indicated that acute consumption of A2 milk results in fewer symptoms and reduced maldigestion in lactose maldigesters [15]. Furthermore, milk containing A1 β-casein undergoes more rapid gastric transit than A2 milk, which suggests less stomach and/or small bowel digestion and more colonic fermentation [16]. Fermentation generates gases including hydrogen, carbon dioxide, and methane in some individuals and causes intestinal distention and fecal urgency. Increased hydrogen in breath samples serves as a valid measure of lactose maldigestion [17,18].

We conducted a double-blind, randomized, crossover trial to investigate whether symptoms, maldigestion, and systemic inflammation persist with chronic A1/A2 as compared to A2 milk consumption. The objective of this study was to examine whether lactose maldigesters adapted to daily consumption of milk containing A1/A2 β-casein for two weeks resulting in fewer symptoms, reduced maldigestion, and lower serum inflammatory markers compared to milk containing only A2 β-casein.

## 2. Materials and Methods

### 2.1. Participant Recruitment and Eligibility Criteria

Four hundred and forty-four individuals expressed interest in participating in the study either via email or telephone (Figure 1). Of these, 109 subjects successfully underwent phone screening and were classified as eligible or ineligible (Appendix B). Eligible participants provided informed consent. We gathered essential demographic data, information on current medication usage, and height and weight measurements for the computation of body mass index (BMI). Participants were assigned a unique identification (ID) number after signing the informed consent. Randomization of subjects was carried out using Research Randomizer (randomizer.com). Randomized milk treatments information was sent to the staff in our Clinical Research Center (CRC), and the CRC staff poured milk in blinded plain cartons labeled with subject ID numbers. Participants and study personnel responsible for collecting data and evaluating outcomes during the 6 h assessments were blinded to the specific milks fed.

Inclusion criteria included not consuming dairy for a minimum of one month prior to screening and avoiding dairy products and all dairy intolerance treatments or products (e.g., Lactaid^®^ dietary supplements; McNeil Nutritionals, LLC, Ft. Washington, PA, USA) for the entire duration of the trial. Lactose maldigestion was identified by an increase of 20 parts per million (ppm) in hydrogen levels following a challenge dose of A1/A2 milk containing 0.5 g of lactose per kg of body weight during a six-hour hydrogen breath test (HBT) (Appendix B).

Exclusion criteria included a documented allergy to milk, pregnancy or lactation; smoking of cigarettes or the use of tobacco or nicotine-containing products within 3 months prior to screening; diagnosis of abnormal gastrointestinal motility; a history of gastrointestinal surgery; the presence of any medical condition with symptoms that could confound data collection related to adverse events, ulcers, diabetes mellitus, congestive heart failure, infection with HIV, hepatitis B, or hepatitis C viruses; a BMI exceeding 35 kg/m^2^; use of dairy intolerance treatment products within 7 days of screening; utilization of antacids and/or proton pump inhibitors; administration of antibiotics or colonic enemas within 30 days preceding screening; concurrent diseases or symptoms that could interfere with the assessment of the cardinal symptoms of dairy intolerance; consumption of ethanol (alcohol) or drug abuse within the past month; chemotherapy; or the use of any investigational drug or participation in an investigational study within 30 days before screening (Appendix B).

Symptoms collected during the study included abdominal pain, bloating, flatulence, diarrhea and fecal urgency. Subjects rated each symptom using a six-point Likert scale, ranging from 0 (indicating no symptoms) to 5 (indicating severe symptoms), with intermediate values representing varying degrees of symptom severity (1 for slight, 2 for mild, 3 for moderate, and 4 for moderately severe).

We enrolled 16 verified lactose maldigesters, from 18 to 65 years old, into a randomized, double-blind, crossover trial. Recruitment of subjects included the distribution of flyers and advertisements in local (West Lafayette, IN, USA) and campus (Purdue University, West Lafayette, IN, USA) newspapers, Purdue student organization newsletters, along with outreach via Purdue campus email. Recruitment began in January 2022, and data collection continued until October 2023.

### 2.2. Study Procedure

The study included two milk interventions: A1/A2 milk, containing 75% A1 β-casein and 25% A2 β-casein (Kroger^®^ 2% reduced fat; The Kroger Co., Indianapolis, IN, USA), and A2 milk, containing 100% A2 β-casein with no A1 β-casein (The a2 Milk Company, Boulder, CO, USA). Both milk products were purchased from Payless (West Lafayette, IN, USA) or Fresh Thyme (West Lafayette, IN, USA). Milk was purchased every two weeks and fed prior to printed best use expiration dates.

We conducted a double-blinded, randomized, crossover study, consisting of two two-week phases, with a minimum six-day wash-out period between phases. Subjects collected milk cartons for two-week milk consumption on the first day of each study phase. Blood samples were collected on the first day of each phase, followed by the daily consumption of 250 mL of milk twice a day along with two meals during a 14-day period. There was at least a 2 h gap between the two meals. Subjects recorded their daily dietary intake and stool frequency during the two-week milk consumption period and rated symptoms abdominal pain, bloating, flatulence, diarrhea, and fecal urgency every day. Reminder emails to consume milk were sent to the subjects every day during the two-week phases, and subjects confirmed milk consumption by responding to the reminder emails. On the 15th day, after two weeks of milk consumption, subjects underwent an HBT following a 12 h fast. The HBT on day 15 of the study phase involved a challenge with the same milk that subjects had consumed for 14 days during the study phase (0.5 g of lactose/kg body weight). Subjects consumed a low-fiber dinner before the test and then fasted. A baseline breath sample was collected at the start of the visit. Blood was drawn at baseline before milk challenge and at 1, 2, 3 h time points after the milk challenge. Breath samples were collected, and subjects rated their symptoms at 0.5, 1, 1.5, 2, 3, 4, 5, and 6 h post-milk ingestion. Hydrogen levels in breath samples, serum markers, including IgG, IgG1, hs-CRP, IL-4, and the antioxidant glutathione (GSH), were analyzed.

### 2.3. Nutrient Analysis

Total sugar, fat, and protein in conventional milk and A2 milk were analyzed by Eurofins Food Integrity and Innovation (Eurofins Food Chemistry Testing US, Inc., Madison, WI, USA). A1 and A2 β-casein in the two types of milk were analyzed using mass spectrometry at Purdue Proteomics Facility. The methodology and results for nutrient analysis have been previously reported [15].

### 2.4. Serum Analysis 

Immunoglobulin G1 (IgG1), glutathione (GSH), and interleukin-4 (IL-4) were measured using ELISA assays as described below. Cobas Integra 400 plus was used for the analysis of high-sensitivity C-reactive protein (hs-CRP) and total immunoglobulin G (IgG). The analyses of GSH and IL-4 were conducted at the Translation Core, Indiana University School of Medicine (635 Barnhill Drive, Indianapolis, IN-46202, USA), while the analyses of IgG1, total IgG, and hs-CRP were carried out at Purdue University (Stone Hall, 700 W State Street, West Lafayette, IN 47906, USA and Bindley Bioscience Center, 1203 W State St, West Lafayette, IN 47906, USA).

#### 2.4.1. IgG1

IgG1 analysis was performed using an Invitrogen Human IgG1 ELISA kit (Thermo Fisher Scientific Inc., Waltham, MA, USA 02451) [19]. Reagents, including wash buffer and assay buffer, were prepared as per the kit manual instructions. Human IgG1 standard dilutions were created with assay buffer, yielding seven standards with different concentrations. Predilution A and Predilution B were prepared by diluting serum samples with assay buffer at a 1:10,000 ratio. The microwell strips were washed with wash buffer, and standards, blank, and Predilution B were added in duplicate. The assay buffer was added to all sample wells. After incubating at room temperature for an hour on a microplate shaker, the strips were washed and aspirated, and the color development was measured at 620 nm using an ELISA reader. The optical density (OD) values were converted into IgG1 concentrations in nanograms per milliliter (ng/mL).

#### 2.4.2. GSH

GSH analysis was carried out using an LSBio GSH ELISA kit (LifeSpan BioSciences, Inc., Seattle, WA, USA 98121) [20]. Standard preparation involved centrifuging and reconstituting the lyophilized standard with sample diluent, followed by seven standard dilutions. Detection reagents A and B were diluted with assay diluents A and B at a 1:100 ratio, respectively. Samples and blanks, along with detection reagent A, were added to the wells, followed by an hour of incubation. After washing, detection reagent B was added, and another 45 min incubation at 37 °C followed. After a final wash, the plate was incubated for 15–30 min with TMB substrate solution, and the reaction was stopped with stop solution. OD values were read at 450 nm.

#### 2.4.3. IL-4

Analysis of IL-4 was conducted using the Human IL-4 immunoassay Quantikine ELISA kit (USA R&D Systems, Inc., Minneapolis, MN, USA 55413) [21]. Six serial dilutions of human IL-4 standard were made by diluting it with calibrator diluent RD5L. In the assay procedure, assay diluent and standards, controls, or samples were added to the wells. The wells were covered and incubated for two hours, followed by washing and incubation with human IL-4 conjugate, and OD values were measured at 450 nm.

#### 2.4.4. Total IgG and hs-CRP

Calibration of the Cobas Integra 400 plus (Block Scientific, Inc., Bellport, NY, USA 11713) was initiated by loading calibrators and controls. An order ID was created for each serum sample to be analyzed, and samples were processed and reviewed for results [22].

### 2.5. Study Ethics 

The study (NCT05669274) was approved by the Purdue Institutional Review Board (IRB-2021-1407) and adhered to the Helsinki Declaration of 1975 as revised in 2008 [23] and the International Conference on Harmonization Good Clinical Practice guidelines [24].

### 2.6. Statistical Analysis

The normality of hydrogen and serum markers was assessed using the Shapiro–Wilk test, and log transformation was applied to correct for non-normality. A two-way repeated measures ANOVA was used to compare hydrogen concentration between A1/A2 milk and A2 milk. Mean concentrations of IgG1, hsCRP, IgG, and GSH in serum after each study phase were calculated, as well as average hydrogen production during the six-hour breath test at the end of each phase. A paired t-test was employed for the comparison of baseline concentrations after the two-week milk consumption between A1/A2 and A2 milk at baseline. The examination of baseline concentrations before and after the two-week milk consumption for both milk types also utilized a paired t-test. In the case of post-challenge serum markers comparing A1/A2 and A2 milk, a two-way repeated measures ANOVA was employed for IgG1, hsCRP, and GSH. However, it was noted that total IgG did not follow a normal distribution for post-challenge concentration. As a result, the Friedman test was used for analysis in this specific case. The total symptom scores for all 14 days were calculated for each symptom by adding the symptom scores for each day. The difference in each symptom between the two milk types was compared using the Wilcoxon signed-rank test. Changes in day-to-day symptoms were analyzed using the cumulative link mixed model. Two-way repeated measures ANOVA and cumulative link mixed model were conducted using R version 4.2.2, paired *t*-tests using Microsoft Excel version 2404, and Wilcoxon signed-rank tests and Friedman tests using IBM SPSS Statistics version 28.0.0.0.

## 3. Results

### 3.1. Participant Baseline Characteristics

Baseline characteristics reported in Table 1 represent the outcomes of the recruitment and selection process. Among the initial 109 subjects assessed for eligibility via phone screening, 21 were found ineligible due to underlying medical conditions or a lack of prior milk avoidance (Figure 1). Of the 88 eligible subjects, 43 participated in the HBT screening, while 45 did not respond to scheduling emails. Eleven subjects did not meet the criteria for lactose maldigestion, and sixteen subjects withdrew from the study after HBT screening. Consequently, 16 lactose maldigesters were randomly assigned to one of the two study phases involving A1/A2 milk and A2 milk. Analysis of symptoms, inflammatory markers, and antioxidants was conducted on all 16 subjects. However, hydrogen analysis was carried out for only 15 subjects due to a misfunction of the hydrogen breath analyzer during one subject’s study phase. All subjects participating in the study resided in the United States during data collection. The study population consisted of 6 males and 10 females, with ages ranging from 19 to 34 years and an average BMI of 23 kg/m^2^. The subjects were ethnically diverse, including 10 Asians, 4 Caucasians, and 1 Asian-Caucasian (mixed race), with 1 subject reporting Hispanic ethnicity (Table 1). One subject did not report the race and ethnicity.

### 3.2. Symptoms 

#### 3.2.1. Daily Symptoms Reported during the Two-Week Milk Consumption

Fecal urgency was significantly reduced during the two weeks of daily consumption of A2 milk compared to A1/A2 milk (*p* = 0.04, *n* = 16). Other symptoms did not exhibit significant differences between the two milk treatments (Figure 2).

The day-to-day analysis revealed that the log-odds of experiencing abdominal discomfort was higher on days 2 (*p* = 0.04) and 10 (*p* = 0.02) compared to day 1 with A1/A2 milk. Log-odds of bloating was higher on day 2 (*p* = 0.02) and lower on days 4 (*p* < 0.01) and 6 (*p* = 0.05) compared to day 1 with A1/A2 milk. The log-odds of experiencing flatulence was higher on day 7 (*p* = 0.01) compared to day 1 with A1/A2 milk, and lower on day 2 (*p* = 0.05) with A2 milk. Log-odds of fecal urgency was higher on days 8 (*p* = 0.02) and 10 (*p* = 0.02) and lower on day 13 (*p* = 0.02) compared to day 1 with A1/A2 milk. Finally, the log-odds of fecal urgency was lower on day 3 (*p* = 0.04) compared to day 1 with A2 milk. All the other days were not different from day 1 for A1/A2 and A2 milk. The results for day-to-day symptom changes are reported in Appendix A.

#### 3.2.2. Symptoms Reported during Milk Challenge HBT after the Two-Week Milk Consumption

Bloating (*p* = 0.03, *n* = 16) and flatulence (*p* = 0.02, *n* = 16) were significantly reduced during the same milk challenge with 0.5 g of lactose/kg body weight following two weeks of daily consumption of A2 milk compared to A1/A2 milk. Other symptoms were not different between the two milk treatments (Figure 3).

### 3.3. Hydrogen

The average hydrogen production by subjects during the six-hour breath test was 57.18 ppm for A1/A2 milk and 59.45 ppm for A2 milk (Figure 4). This difference was not statistically significant (*p* = 0.13, *n* = 15).

### 3.4. Serum Markers 

#### 3.4.1. Baseline Serum Markers: A1/A2 vs. A2

Inflammatory markers, including IgG1, hsCRP, IgG, and the antioxidant GSH, were not different between A2 milk and A1/A2 milk (Table 2). IL-4 was not detected in any of the samples.

#### 3.4.2. Baseline Concentrations: Before vs. after Two-Week Milk Consumption

Inflammatory markers IgG1, hs-CRP, and the antioxidant GSH did not show statistically significant differences between the baseline concentrations on day 1 (before the two-week milk consumption) and day 15 (after the two-week milk consumption) for A2 milk and A1/A2 milk (Table 3). However, the total IgG was different between A1/A2 and A2 milk (*p* = 0.02, *n* = 16). IL-4 was not detected in any of the samples.

#### 3.4.3. Post-Challenge Serum Markers: A1/A2 vs. A2

Concentrations of IgG1, hs-CRP, IgG, and GSH were not different between A1/A2 milk and A2 milk during the first three hours after milk challenge on day 15 (Table 4). IL-4 was not detected in any of the samples.

### 3.5. Adverse Events

No adverse events were reported by any of the subjects.

## 4. Discussion

Daily consumption of A1/A2 milk for two weeks resulted in significantly higher fecal urgency compared to A2 milk. Bloating and flatulence were lower with A2 milk following a same milk challenge dose after the two-week milk consumption compared to A1/A2 milk. However, day-to-day analysis comparing each day’s symptom with the first day of the intervention were not consistently different for both A1/A2 and A2 milk, suggesting no adaptation during the two weeks. Additionally, other symptoms and maldigestion were also not different between A1/A2 and A2 milk. Total IgG concentration was reduced after two-week consumption of A1/A2 milk. However, serum inflammatory markers IgG1, hs-CRP, and antioxidant GSH concentrations did not differ between the two milk types after two weeks of milk consumption.

Our study focused on lactose maldigesters because the primary aim was to investigate the differential effect of A2 and A1/A2 milk on symptoms associated with lactose intolerance, which is most relevant to this specific population. Including a control group of lactose digesters could provide additional context. However, our goal was to directly address the challenges faced by those with lactose maldigestion. Future research could benefit from exploring the effects of these milk types on individuals who digest lactose normally to provide a broader perspective.

We observed substantial variation in serum markers, symptoms, and hydrogen levels. Over 50% of the study population displayed increased IgG1 and hs-CRP serum concentrations after A1/A2 milk consumption and higher GSH levels following A2 milk consumption. Seventy percent of the subjects experienced more symptoms when consuming A1/A2 milk for two weeks compared to A2 milk. However, only four out of fifteen subjects produced more hydrogen after consuming A1/A2 milk than A2 milk. Findings from previous studies suggest that lactose maldigestion is independent of gender, ethnicity, and race but is influenced by the dose and LNP gene expression [25]. Epigenetic factors may also contribute to intolerance and maldigestion. The clinical significance of milk types in lactose maldigesters and the generalizability of results warrant confirmation in larger study populations.

Previously, we reported a more rapid gastric transit of A1/A2 milk than A2 milk in lactose maldigesters [16]. Jianqin and group observed a slower colonic transit of A1/A2 milk compared to A2 milk [13]. These findings suggest that A1/A2 milk moves swiftly through the stomach and retains in the colon for longer. A1/A2 milk likely undergoes less small bowel digestion, leading to prolonged retention in the colon and subsequent colonic fermentation. The increased hydrogen production observed in an acute study with A1/A2 milk in lactose maldigesters compared to A2 milk supports this notion [15]. In Jianqin’s randomized crossover study with two weeks of milk consumption, the colon and overall gastrointestinal transit time were longer with A1/A2 milk than with A2 milk [13]. In our current study, the observed peak hydrogen occurred two hours earlier with A2 milk, aligning with the expected slower transit of A1/A2 milk through the entire gastrointestinal tract compared to A2 milk following a two-week milk consumption.

Immune responses typically emerge within 10 days of antigen exposure [26]. We measured serum inflammatory markers after two weeks of milk consumption. IgG, which has been linked to chronic intestinal inflammation, is detectable within four days of inflammation onset, peaking at two weeks, with a half-life of approximately 25 days [27,28,29]. In contrast, CRP concentration increases rapidly in response to acute inflammation, starting within 4–6 h of inflammation onset and peaking at 36≠50 h, with a half-life of 4–7 h [30,31].

GSH, an antioxidant, is present in various cellular compartments with a turnover time from 4 to 6 days [32,33]. The rate-limiting enzyme in GSH synthesis is cysteine [34]. Exposure to BCM-7, a digestive by-product of A1 β-casein, has been shown to reduce cysteine uptake, and consumption of milk containing A1 β-casein and A2 β-casein led to a decreased plasma GSH concentration compared to milk containing only A2 β-casein [35,36]. A decrease in GSH concentration may increase cellular inflammation, which could reduce disaccharidase activity and lead to lactose maldigestion [37].

The study did not detect IL-4, but a separate study in a Chinese population found significantly lower IL-4 levels following A2 milk consumption compared to A1/A2 milk [13]. These differing IL-4 responses may be attributed to demographic variations.

The major contribution of this study is that it expands upon the findings of the previous study [15] by investigating the effects of prolonged exposure (two weeks) to A1/A2 and A2 milk, whereas the 2020 study focused on the acute effects observed after a single meal. The single milk meal approach highlighted the immediate benefits of A2 milk in reducing abdominal pain and hydrogen production [15]. The current study’s two-week milk consumption provided a more sustained evaluation, showing reductions in bloating, flatulence, and fecal urgency with A2 milk, but without significant differences in hydrogen production. The findings suggest that while acute symptom relief with A2 milk is notable, the benefits for some symptoms become more evident over a longer period.

## 5. Conclusions

In summary, daily consumption of A2 milk for two weeks resulted in significantly lower fecal urgency compared to A1/A2 milk and reduced bloating and flatulence following a lactose challenge, post intervention. Larger sample sizes may yield statistical significance in terms of abdominal pain, hydrogen levels, inflammatory markers, and antioxidant concentrations. The substantial variance observed in hydrogen levels, serum markers, and symptoms calls for further investigation with larger and more diverse study populations, encompassing subjects from various racial and ethnic backgrounds. Future studies could explore how populations accustomed to milk from native breeds respond when introduced to milk from crossbred cows, and vice versa, and examine the gastric transit.

## Figures and Tables

**Figure 1 nutrients-16-01963-f001:**
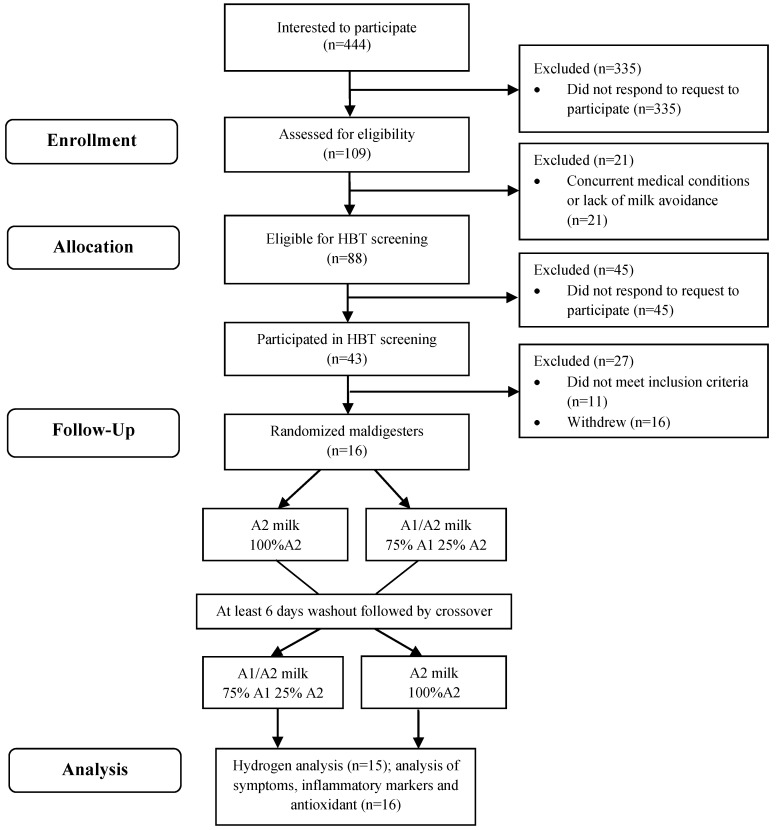
CONSORT diagram. HBT, hydrogen breath test.

**Figure 2 nutrients-16-01963-f002:**
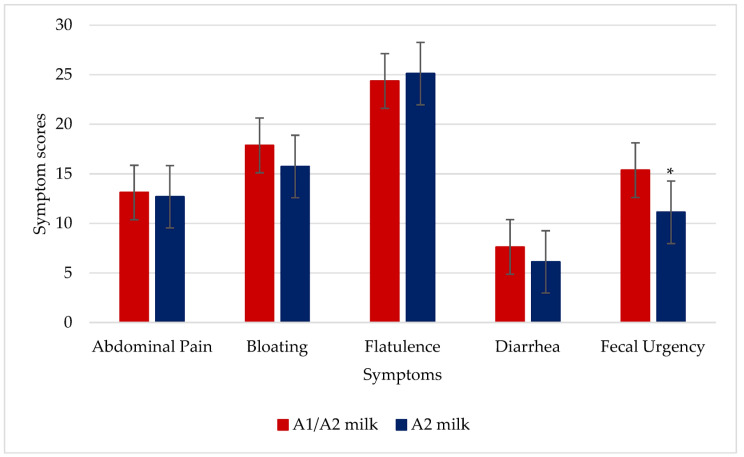
Symptoms produced during the two-week milk consumption of 250 mL of milk per day with two meals in 16 lactose maldigesters. *p* = 0.84 for abdominal pain, *p* = 0.63 for bloating, *p* = 0.98 for flatulence, *p* = 0.33 for diarrhea, and * *p* = 0.04 for fecal urgency due to A2 milk vs. A1/A2 milk.

**Figure 3 nutrients-16-01963-f003:**
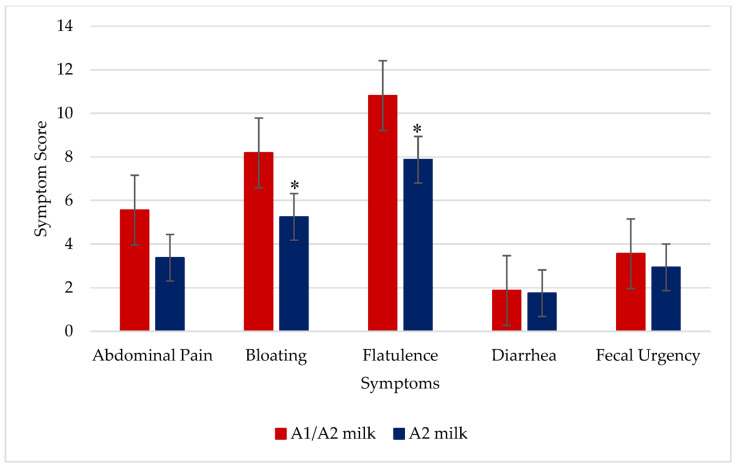
Symptoms produced during a six-hour HBT with a challenge milk dose of 0.5 g of lactose per kg body weight on day 15, after the two-week milk consumption in 16 lactose maldigesters. *p* = 0.14 for abdominal pain, * *p* = 0.03 for bloating, * *p* = 0.02 for flatulence, *p* = 0.79 for diarrhea, and *p* = 0.41 for fecal urgency due to A2 milk vs. A1/A2 milk.

**Figure 4 nutrients-16-01963-f004:**
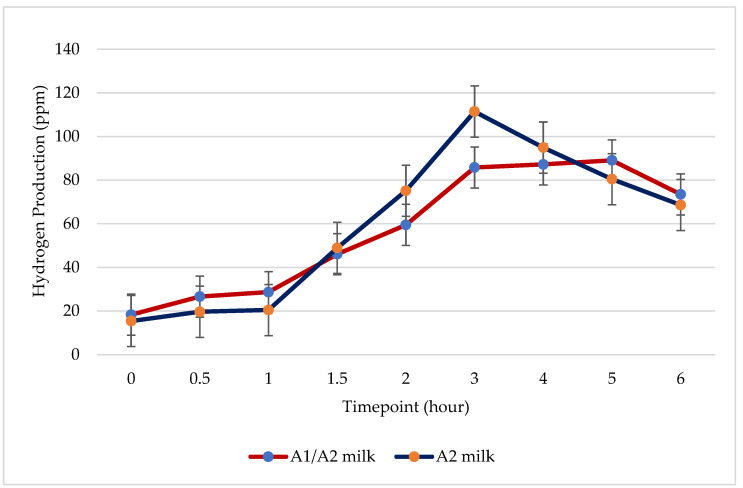
Comparison of hydrogen production between A1/A2 milk and A2 milk during a six-hour HBT with a milk challenge dose of 0.5 g per kg body weight on day 15, following two weeks of milk consumption in 15 lactose maldigesters.

**Table 1 nutrients-16-01963-t001:** Baseline and demographic characteristics.

Age, mean (range); years	25 (19–34)
Bodyweight, mean (range); kg	66 (51–91)
Height, mean (range); cm	168 (152–188)
BMI, mean (range); kg/m^2^	23 (20–27)
Male/female, *n*/*n*	6/10
Asian	10
Caucasian	4
Mixed (Asian and Caucasian)	1
Hispanic	1
Non-Hispanic	9

Baseline and demographic characteristics of lactose maldigesters (*n* = 16) enrolled in this randomized, double-blinded trial comparing A1/A2 milk with A2 milk. BMI, body mass index.

**Table 2 nutrients-16-01963-t002:** Comparison of baseline serum markers concentration after the two-week milk consumption.

Marker	A1/A2 Milk	A2 Milk	*p*-Value
IgG1 (ng/mL)	598.34 ± 108.23	523.13 ± 104.75	0.45
hs-CRP (mg/L)	1.23 ± 0.32	2.27 ± 1.29	0.45
Total IgG (g/L)	10.22 ± 0.61	10.42 ± 0.66	0.47
GSH (ng/mL)	39.66 ± 7.62	42.62 ± 9.45	0.39
IL-4	Not detected	Not detected	Not available

Comparison of serum markers produced after two weeks of A1/A2 milk versus A2 milk consumption using paired *t*-tests. Analyses were conducted on serum samples collected on day 15 following the two-week milk consumption and before the six-hour hydrogen breath test. Data are reported as mean ± standard error (SE).

**Table 3 nutrients-16-01963-t003:** Comparison of baseline serum markers concentration before and after the two-week milk consumption for A1/A2 milk and A2 milk.

Marker	A1/A2 Milk on Day 1	A1/A2 Milk on Day 15	*p*-Value	A2 Milk on Day 1	A2 Milk on Day 15	*p*-Value
IgG1 (ng/mL)	532.52 ± 103.21	598.34 ± 108.23	0.54	493.52 ± 80.01	523.13 ± 104.75	0.55
hs-CRP (mg/L)	1.78 ± 0.61	1.23 ± 0.32	0.41	1.19 ± 0.35	2.27 ± 1.29	0.44
Total IgG (g/L)	10.95 ± 0.64	10.22 ± 0.61	0.02	10.54 ± 0.63	10.42 ± 0.66	0.68
GSH (ng/mL)	43.01 ± 8.28	39.66 ± 7.62	0.13	49.76 ± 12.13	42.62 ± 9.45	0.12
IL-4	Not detected	Not detected	NA	Not detected	Not detected	NA

Comparison of baseline serum marker concentrations before versus after the two-week milk consumption for A1/A2 milk and A2 milk using paired *t*-tests. Analyses were conducted on serum samples collected on day 1 before the two-week milk consumption and on day 15 following the two-week milk consumption but before the hydrogen breath test milk challenge. Data are reported as mean ± standard error (SE). NA, not available.

**Table 4 nutrients-16-01963-t004:** Comparison of serum markers concentration for the first three hours after the milk challenge on day 15.

Marker	A1/A2 Milk	A2 Milk	*p*-Value
IgG1 (ng/mL)	351.78 ± 53.67	294.76 ± 43.35	0.86
hs-CRP (mg/L)	1.21 ± 0.18	2.29 ± 0.73	0.87
Total IgG (g/L)	10.13 ± 0.38	10.19 ± 0.40	0.77
GSH (ng/mL)	35.73 ± 3.40	36.24 ± 3.87	0.69
IL-4	Not detected	Not detected	Not available

Comparison of IgG1, hs-CRP, and GSH concentrations during the first three hours followed by the milk challenge on day 15 for A1/A2 milk versus A2 milk using two-way repeated measures ANOVA and total IgG using the Friedman test. Analyses were conducted with serum samples collected at 1, 2, and 3 h time points on day 15 following the milk challenge dose of 0.5 g of lactose/kg bodyweight for the six-hour hydrogen breath test. Data are reported as mean ± standard error (SE).

## Data Availability

The data presented in this study are available on request from the corresponding author. The data are not publicly available due to privacy and ethical restrictions.

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
