# Peer review of "Prolonged Consumption of A2 β-Casein Milk Reduces Symptoms Compared to A1 and A2 β-Casein Milk in Lactose Maldigesters: A Two-Week Adaptation Study"

_nutrients, 2024, doi:10.3390/nu16121963_

Round 1

Reviewer 1 Report

Comments and Suggestions for Authors

See attached.

Author Response

Thank you very much for the comments! We appreciate your feedback. We have  highlighted the changes made in the updated manuscript.

  1. Changed the title to: Prolonged Consumption of A2 β-Casein Milk Reduces Symptoms Compared to A1 and A2 β-Casein Milk in Lactose Maldigesters: A Two-Week Adaptation Study
  2. Thanks for your comment on BCM-7! It has been helpful to learn why we could not find a difference in BCM-7. We have removed all methods and results for BCM-7 in the manuscript.
  3. We have removed the sentence about the transit in small bowel and included only the colonic transit result from Jianqin's paper. The rest of the paragraph has been modified accordingly.
  4. A paragraph has been added to the end of the discussion section.
  5. Added ‘future studies on gastric transit’ part to the conclusion section.

Reviewer 2 Report

Comments and Suggestions for Authors

The paper “A two-week adaptation period with milk containing both A1 and A2 β-casein does not improve lactose digestion or tolerance to match the tolerance to A2 β-casein milk” is interesting, and also has very important aims for the human health. Knowing in fact that A1 β-casein, containing histidine, causes the release of the pro-inflammatory compound betacasomorphin-7 (BCM-7) during digestion, whereas A2 β-casein, with proline, does not release BCM-7; and also knowing that previous studies have shown that milk containing A1 β-casein leads to more pronounced lactose intolerance symptoms; they wanted to verify whether this was true by comparing the effects, for lactose maldigesters, of a diet containing only type A2 β-casein milk, with a diet containing mixed A1/A2 milk.

The authors conducted the Participant Recruitment and eligibility criteria in a correct way, selecting, at the end of the process, a group of 16 subjects, all lactose maldigesters, to be divided into two subgroups: one was given A2 milk, while the others were given A1/A2 milk.

However, although it is clear upon careful reading, that the choice of participants in the experiment was well conducted, in practice, in reality it is not described very clearly in the manuscript, and understanding requires a little patience. For example, the first paragraph immediately states that 16 subjects were involved in the experimentation; while in the following paragraph we start again to describe the entire selection process starting from 444 subjects, with all the subsequent elimination phases. The reader wonders what those 16 subjects have to do with these 444. Only at the end does it become clear that the 16 are the final result of the entire selection process. But then we must not declare them immediately at the beginning, but respect, in the logic of the explanation, the temporal process with which they were selected; so, this paragraph (page 2 lines 59-64) should be moved to the end, to avoid confusion. Even the selection, with all the exclusion criteria, described graphically in Figure 1 and also schematised in Appendix A, should also be better detailed in the explanation in the text, citing these two descriptions more often in the text, the schematic one in Appendix A and the one graphic of Figure 1, to make reading this paragraph less difficult for the reader.

Further question: I understand that your aim is to highlight the differences between the symptoms generated by A2 milk, on the one hand, and A1/A2 milk, on the other, ONLY on lactose maldigesters, but it would not have been useful, as a control group , also highlight the effects of both milks on people who digest lactose normally? In this way, the similarities and differences between the two experimental groups could be better highlighted.

Overall, therefore, being so interesting, so useful in an applicative sense and for human health, and due to the large amount of analyses and results, it is a research work of considerable importance and relevance. Some aspects, including those described above, can be easily modified, making the exposition clearer, and are not of a substantial nature. The paper, therefore, in my opinion, requires a minor revision.

Detailed comments:

Page 2, the aim, lines 52-56: The sentence with the objective is too long and difficult to understand, and should be reformulated.

Page 5: isn't paragraph 3.1 (Participant Baseline Characteristics) still a part that should be put in Material and Methods? If you want to leave it in Results, the reason should be justified with a sentence.

Page 6 line 229: “Their”? It is only one, so it is correct to write “its”.

Page 8 lines 288-289: “However, IgG was different between A1/A2 and A2 milk (p=0.02, n=16)” In which Table I can find this data? I don’t find it nor in Table 2 neither in Table 3. It is better to avoid to describe in the text data that are not present in a Table. Or, alternatively, add them in the Table.

Page 10 lines 335-336: “These findings… bowel”; the sentence is suspended, please reformulate it.

Detailed comments (of editorial nature)

Obviously these comment are much less important than those above.

 Be careful to the rules for Title: I think that all names must be in Capital letters; moreover, it is a little long, you can not shorten it?

“ml” must be written “mL” everywhere. Check it.

Page 2 line 90: “kg/m2”: the “2” I think should be put at the superscript.

Pages 4-9: The paragraphs are all attached to each other, while you should leave a blank line between one paragraph and another, and also between the text and the tables or figures.

In the square bracket of citations, when you have two or more numbers, they must be written without blank space between them. I mean [1,2] is correct, while [1, 2] is not correct.

Comments on the Quality of English Language

Some sentences should be reformulated, but overall English is good.

Author Response

Thank you very much for the comments! We appreciate your feedback. We have highlighted the changes made in the updated manuscript.

Responses from the authors:

  1. Moved lines 59 to 64 to the end of the section. It is the last paragraph under 'Participant Recruitment and Eligibility Criteria'
  2. Cited relevant sentences in the Methods and Results section with Appendix A and Figure 1 to make it easier for the reader to understand.
  3. Thanks for the comment. Our study focused exclusively on lactose maldigesters because the primary aim was to investigate the differential effect of A2 and A1/A2 milk on symptoms associated with lactose intolerance, which is most relevant to this specific population. Including a control group of lactose digesters could provide additional context; however, our goal was to directly address the unique challenges faced by those with lactose maldigestion. Future research could benefit from exploring the effects of these milk types on individuals who digest lactose normally to provide a broader perspective. - Added this as the second paragraph in discussion section.

Detailed comments:

  1. Page 2, the aim, lines 52-56: Edited the objective to make it more digestible.
  2. Page 5: Added this sentence in the results section for justification: Baseline characteristics reported in Table 1 represent the outcomes of the recruitment and selection process.
  3. Page 6 line 229: Corrected the sentence.
  4. Page 8 lines 288-289: Row 3, column 4 in Table 3 has the result for Total IgG (p=0.02). We have added the word 'total' to the sentence to avoid confusion.
  5. Page 10 lines 335-336: Paragraph rewritten following suggestions from the reviewer and corrected the sentence in the process.

Detailed comments (of editorial nature):

  1. Changed the title following Nutrients rules.
  2. Replaced "ml" with "mL" throughout the manuscript.
  3. Page 2 line 90: Changed '2' in 'm2' to superscript.
  4. Pages 4-9: Added spaces between one paragraph and another, and also between the text and the tables or figures.
  5. Removed space between the numbers in citation.